# Learning Mobile Robot Navigation in the Dense Crowd with Deep Reinforcement Learning

**Keyu Li**
Department of Electronic Engineering
The Chinese University of Hong Kong
Shatin, Hong Kong
kyli@link.cuhk.edu.hk

**Ye Lu**
Department of Electronic Engineering
The Chinese University of Hong Kong
Shatin, Hong Kong
luyyy@link.cuhk.edu.hk

## Abstract

In recent years, the growing demand for more intelligent service robots is pushing the development of mobile robot navigation algorithms. Moving in a dense crowd safely and efficiently is an important yet challenging task for the operation of the mobile robots. Reinforcement learning (RL) approaches have shown superior ability in solving sequential decision making problems, and recent work has explored its potential to learn navigation polices in a socially compliant manner. In our project, we propose to apply an RL framework to develop a smart system for human-aware navigation. We propose to use value function approximation based methods to learn human-aware navigation policies and implement the reward shaping, hindsight experience replay, and curriculum learning techniques for our task. The effectiveness of our methods is validated in a simulated human-aware navigation environment. The video demonstration is at `https://mycuhk-my.sharepoint.com/:v:/g/personal/1155131468_link_cuhk_edu_hk/ERJ6SdHXiVZNrT2FiyapPaEBYg0RxWedc2h6_tM-X1q0iQ?e=q2recC`.

## 1 Introduction

With the dramatic development of machine intelligence in recent years, robots are expected to be able to work in social space shared with humans, just like in the real-world environment. Navigation in a dense crowd socially is necessary and requires robot have the ability to understand human behavior and take actions under their cooperative rules, which is a challenge task. Early works divide the task into two separate steps, prediction and planing, which could cause the freezing robot problem (10). Reinforcement learning (RL) framework, as a promising alternative, has been used to learn optimal navigation policies so as to encode the underlying interactions among agents to improve the performance. Some significant progress has been made in recent works. Here, we propose to study the performance of different deep RL methods to tackle the human-aware navigation problem based on a simulation environment built in (2).

## 2 Related Work

**Hand-crafted interaction models.** In early works, some well-designed interaction models are applied to help robots understand social behaviors and avoid obstacles in navigation. These models commonly rely on some hand-crafted functions and are limited to some specific scenarios. One representative work is the Social Force (SF) model (6), which endows a way to analyze crowd behaviors based on interaction forces. It has been extensively used in robotics for both simulation and real-world environments. Interacting Gaussian processes (IGP) model (10) is another way to estimate crowd interaction from data in navigation, describing a probabilistic interaction between multiple navigating entities.

**Imitation learning approaches.** In another kind of works, imitation learning algorithms are utilized to make agents learn optimal navigation policies from expert demonstrations. By directly mimicking the provided demonstrations, navigation polices can be developed to control actions based on different inputs, which is named as behavioral cloning (9) method. Additionally, inverse reinforcement learning (IRL) (7) is another widely used method in imitation learning, learning a mapping from perceptual features to costs to achieve the expert demonstrated behavior. The main challenge for these methods is the scale and quality of expert demonstrations, which is resource consuming and limited by human efforts.

**Reinforcement learning methods.** In recent years, deep reinforcement learning (RL) methods have been intensively investigated and achieved rapid development in various fields, as well as robot navigation. In crowd-aware robot navigation, there have been some works (4)(3)(5) using RL models to learn optimal policies with agent state information in a social cooperative way, implicitly encoding the interactions and cooperation among agents. However, early existing models shares some limitations, modeling the crowd impact by a simplified aggregation of the pairwise interactions and focusing on one-way interactions from humans to the robot. These limitations degrade the performance of cooperative planning in complex and crowded scenes. To overcome these limitations, a recent study (2) propose to rethink human-robot pairwise interactions and jointly model human-robot and human-human interactions in a RL framework, outperforming the earlier RL methods.

## 3 Problem Formulation

Suppose there is a human-robot coexisting environment where a robot and $n$ humans are navigating in a 2D workspace denoted as the X-Y plane. The humans and the robot have their own destinations, and the goal of the robot is to navigate to its destination without colliding with humans. We can formulate this problem as a sequential Markov Decision Process (MDP) model in the framework of reinforcement learning. It is assumed that humans do not avoid the robot during the navigation, and each agent can be simplified as a moving circle. We assume that the position, velocity and radius of each agent can be observed by other agents, which are denoted as $\boldsymbol{p} = [p_x, p_y]$, $\boldsymbol{v} = [v_x, v_y]$ and $r$, respectively. The goal position $\boldsymbol{g} = [g_x, g_y]$ and preferred speed $v_{pref}$ cannot be observed by other agents. A robot-centric frame defined in (4) is adopted here to make the state representation more general and versatile, where the origin of the frame is set at the current position of the robot at time $t$ $\boldsymbol{p}_t$, and X-axis points at its goal position $\boldsymbol{g}$. Let $d_g$ denote the distance from $\boldsymbol{p}_t$ to $\boldsymbol{g}$, and $d^i$ denote the distance from $\boldsymbol{p}_t$ to the position of the $i$-th human at time $t$ $\boldsymbol{p}_t^i$. After transformation, the state of the robot at time $t$ $\boldsymbol{S}_t$ and the observable state of the $i$-th human at time $t$ $\boldsymbol{O}_t^i$ become:

$$
\begin{aligned}
\boldsymbol{S}_t &= [d_g, v_{pref}, v_x, v_y, r], \\
\boldsymbol{O}_t^i &= [d^i, p_x^i, p_y^i, v_x^i, v_y^i, r^i, r^i + r].
\end{aligned}
\tag{1}
$$

The joint state of all $(n+1)$ agents at time $t$ can be obtained by concatenating the state of the robot with the observable states of humans as $\boldsymbol{J}_t = [\boldsymbol{S}_t, \boldsymbol{O}_t^1, \boldsymbol{O}_t^2, \ldots, \boldsymbol{O}_t^n]$. Assume that the robot can change its velocity immediately according to the action command at time $t$ $\boldsymbol{a}_t$, which is determined by the navigation policy: $\boldsymbol{v}_t = \boldsymbol{a}_t = \boldsymbol{\pi}(\boldsymbol{J}_t)$. The corresponding reward at time $t$ is denoted as $R(\boldsymbol{J}_t, \boldsymbol{a}_t)$. The definition of reward function in (4)(3)(2) is shown in (2), where $d_{min}$ is the shortest separation distance between the robot and humans within the decision interval $\Delta t$, and $d_c$ represents the minimum comfortable distance that humans can tolerate.

$$
R(\boldsymbol{J}_t, \boldsymbol{a}_t) = \begin{cases} -0.25, & \text{if } d_{min} < 0; \\ 0.5 * (d_{min} - d_c), & \text{if } 0 < d_{min} < d_c; \\ 1, & \text{if } d_g = 0; \\ 0, & \text{otherwise.} \end{cases}
\tag{2}
$$

Equation (2) indicates that actions that lead the robot to the goal will be awarded while actions that may cause collision or discomfort to pedestrians will be penalized. If the robot satisfies any of the following conditions: a) reaches the destination (i.e., the position is close enough to the goal), b) collided with a human, and c) the navigation time exceeds the time limit, the episode will be terminated.

With an optimal navigation policy $\boldsymbol{\pi}^\star(\boldsymbol{J}_t)$, the optimal value of the joint state $\boldsymbol{J}_t$ at time $t$ can be formulated as:

$$V^\star(\boldsymbol{J}_t) = \sum_{i=0}^{K} \gamma^{i \cdot \Delta t \cdot v_{pref}} \cdot R(\boldsymbol{J}_t, \boldsymbol{a}_t^\star),\tag{3}$$

where $K$ is the total number of decision steps from the state at time $t$ to the final state, $\Delta t$ is the decision interval between two actions, and $\gamma \in (0, 1)$ is a discount factor in which the preferred speed $v_{pref}$ is introduced as a normalization parameter for numerical reasons (4). The optimal policy is formulated by maximizing the cumulative reward as the following:

$$\begin{aligned} \boldsymbol{\pi}^\star(\boldsymbol{J}_t) = \underset{\boldsymbol{a}_t \in \boldsymbol{A}}{\arg\max}\, & R(\boldsymbol{J}_t, \boldsymbol{a}_t) + \gamma^{\Delta t \cdot v_{pref}} \cdot \\ & \int_{\boldsymbol{J}_{t+\Delta t}} P(\boldsymbol{J}_{t+\Delta t} \mid \boldsymbol{J}_t,\, \boldsymbol{a}_t) \cdot V^\star(\boldsymbol{J}_{t+\Delta t})\, \mathrm{d}\boldsymbol{J}_{t+\Delta t}, \end{aligned}\tag{4}$$

where $\boldsymbol{A}$ is the action space (i.e., the set of velocities that can be achieved), $P(\boldsymbol{J}_{t+\Delta t} \mid \boldsymbol{J}_t,\, \boldsymbol{a}_t)$ is the transition probability from $\boldsymbol{J}_t$ to $\boldsymbol{J}_{t+\Delta t}$ when action $\boldsymbol{a}_t$ is carried out. This probability describes the uncertainty of the next joint state due to the unknown crowd behavior.

## 4 Methods

### 4.1 Background

#### 4.1.1 Value Function Approximation (VFA)

In our study, we adopt the Value Function Approximation (VFA) algorithm as the basic RL algorithm to solve our task. In detail, we use a deep neural network with the architecture proposed in (2) as the Value Function Approximator to approximate the optimal value function $\hat{V}(\boldsymbol{J}_t)$. In the policy evaluation, we follow the idea in previous methods (3)(4)(2) that the dynamics of the environment can be considered accessible, i.e., the next joint state of the robot and humans $\boldsymbol{J}_{t+1}$ after the robot takes an action at the current state $\boldsymbol{J}_t$ can be directly assessed by the robot (2) or precisely approximated using some predictive models (8): $\boldsymbol{J}_{t+1} \leftarrow propogate(\boldsymbol{J}_t, a_t)$. Then the optimal policy can be retrieved from the optimal value function $V^\star(\boldsymbol{J}_{t+\Delta t})$. By this simplification, the calculation of the optimal policy in (4) can be modified as (5).

$$\boldsymbol{\pi}^\star(\boldsymbol{J}_t) = \underset{\boldsymbol{a}_t \in \boldsymbol{A}}{\arg\max}\, R(\boldsymbol{J}_t, \boldsymbol{a}_t) + \gamma^{\Delta t \cdot v_{pref}} \cdot \hat{V}(\boldsymbol{J}_{t+\Delta t}).\tag{5}$$

For example, the pedestrians can be assumed to move with constant velocities during the time interval $[t, t + \Delta t]$ since the duration $\Delta t$ is very small. Therefore, by using a constant velocity model (CVM), the next joint state of robot and humans can be predicted: $\boldsymbol{J}_{t+\Delta t} \leftarrow \mathrm{CVM}(\boldsymbol{J}_t, \Delta t, \boldsymbol{a}_t)$.

The value network is trained by the temporal-difference (TD) method with experience replay and fixed target network techniques.

#### 4.1.2 Hindsight Experience Replay (HER)

In our problem formulation, the defined reward function (2) is sparse, where the rewards are usually uneasy to reach with random explorations. Dealing with sparse rewards is always more challenging in RL. Hindsight Experience Replay (HER) proposed in (1) is a popular method to solve such problems. The key insight of HER is that the agent can also learn useful information from the failed rollouts by viewing the final state as its additional goal, as if the agent intended on reaching this state from the very beginning. The idea is realized by relabeling the failed rollouts as successful ones. The details of HER algorithm for VFA learning in our task is outlined in Algorithm 1. The line 14-25 is the HER part and the left part is the standard VFA learning algorithm.

For each episode the agent experiences: when there is a collision or the agent achieves the original goal, we directly store it in the experience replay buffer; when it is a timeout situation without discomfort, we relabel the final state as the goal and the last reward as a successful one and then store the modified trajectory in the replay buffer. HER is a simple method without complicated reward engineering and can help improve the sample efficiency in RL.

**Algorithm 1** VFA Learning with HER

---

1: Initialize value network $V$ and target value network $\hat{V}$
2: Initialize experience replay memory $E$
3: **for** episode $i \in [1, M]$ **do**
4:     Sample an initial state $\boldsymbol{J}_0$ with the original goal $\boldsymbol{g}$
5:     **for** $t = 0\,, T - 1$ **do**
6:         $\boldsymbol{a}_t \leftarrow \boldsymbol{\pi}^\star(\boldsymbol{J}_t) = \arg\max_{\boldsymbol{a}_t \in \boldsymbol{A}} R(\boldsymbol{J}_t, \boldsymbol{a}_t) + \gamma^{\Delta t \cdot v_{pref}} \cdot \hat{V}(\boldsymbol{J}_{t+1})$
7:         Execute the action $\boldsymbol{a}_t$ and observe a new state $\boldsymbol{J}_{t+1}$
8:     **end for**
9:     Record information *info* of the last state $\boldsymbol{J}_T$
10:    **if** *info = ReachGoal* or *Collision* **then**
11:        **for** $t = 0\,, T - 1$ **do**
12:            Store the transition $(\boldsymbol{J}_t, \boldsymbol{a}_t, \boldsymbol{r}_t, \boldsymbol{J}_{t+1})$ in $E$
13:        **end for**
14:    **else if** *info = Timeout* **then**
15:        Relabel the final agent position as the addition goal:
16:        $\boldsymbol{g}' \leftarrow \boldsymbol{p}_T$
17:        **for** $t = 0\,, T - 1$ **do**
18:            Obtain $\boldsymbol{J'}_t$ and $\boldsymbol{J'}_{t+1}$ with $\boldsymbol{g}'$
19:            **if** $\boldsymbol{p}_t = \boldsymbol{g}'$ **then**
20:                $\boldsymbol{r'}_t = 1$
21:            **else**
22:                $\boldsymbol{r'}_t = \boldsymbol{r}_t$
23:            **end if**
24:            Store the transition $(\boldsymbol{J'}_t, \boldsymbol{a}_t, \boldsymbol{r'}_t, \boldsymbol{J'}_{t+1})$ in $E$
25:        **end for**
26:    **end if**
27:    **for** $t = 0\,, N$ **do**
28:        Sample a minibatch $B$ from the replay buffer $E$
29:        Set target $y_i = \boldsymbol{r}_i + \gamma^{\Delta t \cdot v_{pref}} \cdot \hat{V}(\boldsymbol{J}_{i+1})$
30:        Update value network $V$ by gradient descent
31:    **end for**
32:    **if** episode % target update interval $= 0$ **then**
33:        Update target network $V \leftarrow \hat{V}$
34:    **end if**
35: **end for**

---

## 4.2 Methods with Expert Demonstrations

Expert demonstrations are commonly utilized in previous studies of crowd-aware robot navigation, which can be used as a guide for the optimal RL policy exploration and to accelerate the learning process. The application of the additional domain expertise is also as an alternative to solve the sparse reward problem. Therefore, we first study the performance of different methods under the condition of having expert demonstrations.

Firstly, we implement the standard VFA learning with just initializing the experience replay memory with the expert demonstrations. It functions as a baseline for the methods with demonstration data. Secondly, we study the VFA learning combined with imitation learning (IL), which is a popular practice with demonstration data. Specifically, the value network is also initialized with demonstrations through supervised learning on the demonstration state-action pairs. Thirdly, we extend the VFA learning with HER algorithm, boosting the training process with demonstrations. It is realized by initializing the experience replay memory in the algorithm of VFA learning with HER (Algorithm 1).

By designing the three contrast experiments, we want to study the separate contributions of IL and HER to the standard RL algorithm VFA learning. It is our premier experimental work to explore possible methods for our task.

### 4.3 Methods without Expert Demonstrations

In spite of the benefits using expert demonstrations for training of the agent, there are some limitations in practice. First of all, the quantity and quality of expert demonstrations have critical influence on the performance of the agent in environments with sparse rewards. It is resource-consuming and difficult to collect a large dataset of high-quality human demonstration data. In addition, we may even have no access to reliable domain expertise for RL in the real world problems. In our implementation, we collect demonstration data by letting the agent act using the policy of simulated humans like (2) did, but this is not applicable in reality and will reduce the difficulty of the agent learning to navigate in the simulation environment. Therefore, in this section we explore the possible methods under the assumption that no expert demonstrations are provided, to develop more versatile algorithms for crowd-aware robot navigation. We propose two solutions to deal with the sparse reward problems without expert demonstrations, i.e. using shaped rewards, and curriculum learning with HER.

#### 4.3.1 Reward shaping

When solving sparse reward tasks, using shaped reward can be beneficial with carefully engineering. Considering the task of robot navigation is to reach some goal, we propose to incorporate an distance-to-goal reward to the original sparse reward (2). Since simple distance-to-goal reward shaping often fails as it renders learning vulnerable to local optima, it needs to be well designed for a successful application. We shape the reward only based on the distance-to-goal function without any additional domain knowledge, as shown in (6):

$$R(\boldsymbol{J}_t, \boldsymbol{a}_t) = \begin{cases} -1, & \text{if } d_{min} < 0; \\ 0.5 * (d_{min} - d_c), & \text{if } 0 < d_{min} < d_c; \\ 2, & \text{if } d_g = 0; \\ -\alpha * dist, & \text{otherwise.} \end{cases} \tag{6}$$

where $dist = \|\boldsymbol{p} - \boldsymbol{g}\|$ is the distance from the agent to the fixed goal and $\alpha$ is a hyper-parameter. In our task, we set $\alpha = 0.002$ considering the relative magnitudes between the distance value and the other reward values and having several experimental trials.

While the pure VFA learning can converge to some successful policies with our shaped reward, its performance is still limited. First, the collision rate can be relatively high while the success rate has been converged. Second, by manually tuning the hyper-parameter $\alpha$, obtaining the optimal policy is engineering cost and unwarrantable.

#### 4.3.2 Curriculum Learning with HER

Another method we use to deal with the environment with sparse reward is to implement the HER technique, which was originally designed for such tasks. As illustrated in Algorithm 1, we relabelled the final state of the "Timeout" trajectories as goals to store in the experience replay. Therefore, the agent can also learn to navigation with the failed experiences.

In practice, we find that when there are many humans in the simulated environment and the punishment for collision is high, the agent can hardly get successful trajectories during the training, even equipped with the HER technique. Therefore, we introduce the Curriculum Learning (CL) in our method, to make the agent start out with easy tasks and then gradually increase the task difficulty. In our implementation, we first train the agent in a simple environment that contains only one human to train the agent to navigate without collision, and then transfer it to a more complex environment with 5 humans.

## 5 Experiments

### 5.1 Simulation Environment

The simulation environment for human-aware navigation is built in Python. There are 5 humans in the simulated crowd, whose navigation policy is determined by the optimal reciprocal collision avoidance (ORCA) method (11). We use circle crossing scenarios, where the positions of the humans and the robot are randomly initialized on a circle of radius $4m$ and their goal positions are on the

opposite side of the circle. We assume holonomic kinematics for the robot, i.e., it can move in any direction. The action space consists of 9 discrete actions in total: the stop action and the speed of $1m/s$ with 8 heading directions evenly spaced between $[0, 2\pi)$.

## 5.2 Experimental Results with Expert Demonstrations

In the experiments with expert demonstrations, the demonstration data are generated by navigating with ORCA policy for 3000 episodes, and used to initialize the experience replay memory of the agent. For imitation learning, we pre-train the value network for 50 epochs on the demonstration data before the reinforcement learning. For reinforcement learning, we apply decayed learning rate and $\varepsilon$-greedy policy exploration. The learning rate is set optimal for each model separately. The exploration rate of the $\varepsilon$-greedy policy decays linearly from 0.5 to 0.1 in the first 5000 episodes and stays 0.1 for the remaining episodes. The value network is trained for 10000 RL episodes. The training set and validation set use different seeds for initialization of the positions of the agents and simulated humans.

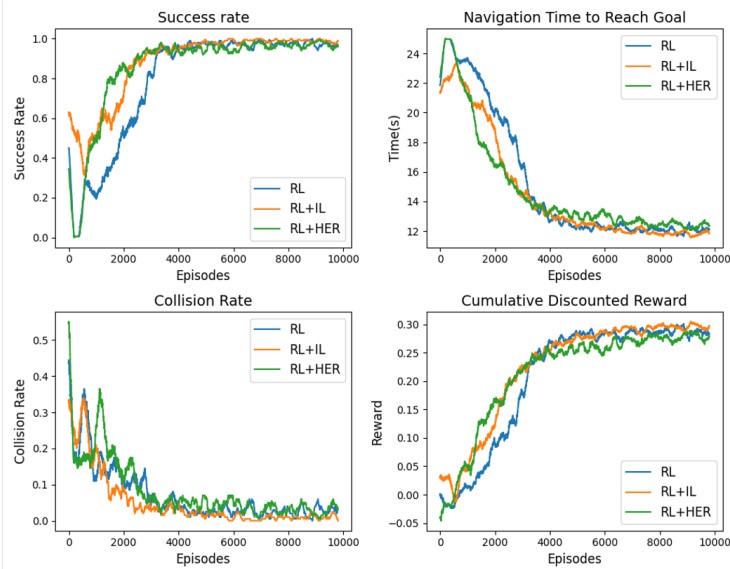

Figure 1: The comparison of three methods with demonstrations

The results among the standard VFA model, the VFA model with IL and the VFA model with HER are shown in Fig. 1. It can be seen that all of the three methods can obtain a success rate close to 100 percent with relatively low collision, which shows the importance of expert demonstrations. In addition, the method with IL achieves the highest reward, presenting the benefit of pretraining the network with IL. And the method with HER converges the fast, proving that HER can help improve the sample efficiency and accelerate the learning process.

## 5.3 Experimental Results without Expert Demonstrations

In the experiments Without demonstration data, we initialize the experience replay memory with 3000 episodes generated with random actions. Other implementation details are the same as those with expert demonstrations.

We first evaluate the method using our shaped reward. The learning curves are shown in Fig. 2. It can be seen that for both training and validation sets, the cumulative discounted reward gradually increases during the training process, and the success rate and navigation time of the agent improve a lot. This means the method using shaped reward can successfully learn the navigation actions to reach the goal in the crowd-navigation scenarios. However, we observe that the collision rate first decreases in the first 5000 training episodes, but increases in the remaining 5000 episodes, which is undesirable. We think it is related to our self-designed reward function, which requires more carefully engineering.

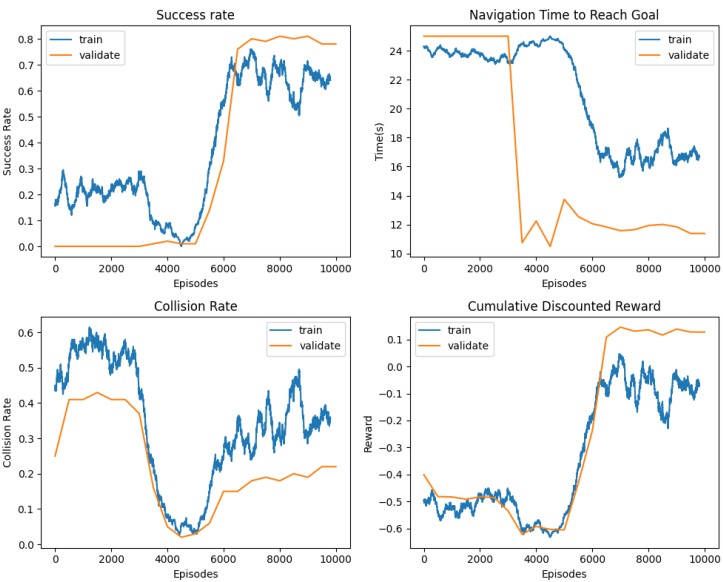

Figure 2: Learning curves of the reward shaping method without using demonstrations.

Then, we evaluate our method using CL with HER. Firstly, we compare the methods with and without using HER for the navigation task in an environment with only 1 human. Fig. 3 shows the results, where (a) is for the method without HER and (b) is for the method with HER. We can see that the pure VFA learning without HER fails as the agent can never get successful experiences. This is because the reward is too sparse and there are no demonstrations as a guidance. Hence, it is difficult to learn a successful navigation policy even in the environment of only 1 human. However, with the help of HER, the experience replay memory becomes more informative and enables the successful learning process, as shown in Fig. 3 (b). During training, the success rate gradually increases and converges to nearly 100% with a low collision rate. The experimental result proves the effectiveness of HER algorithm in dealing with sparse reward problems, without any additional domain expertise.

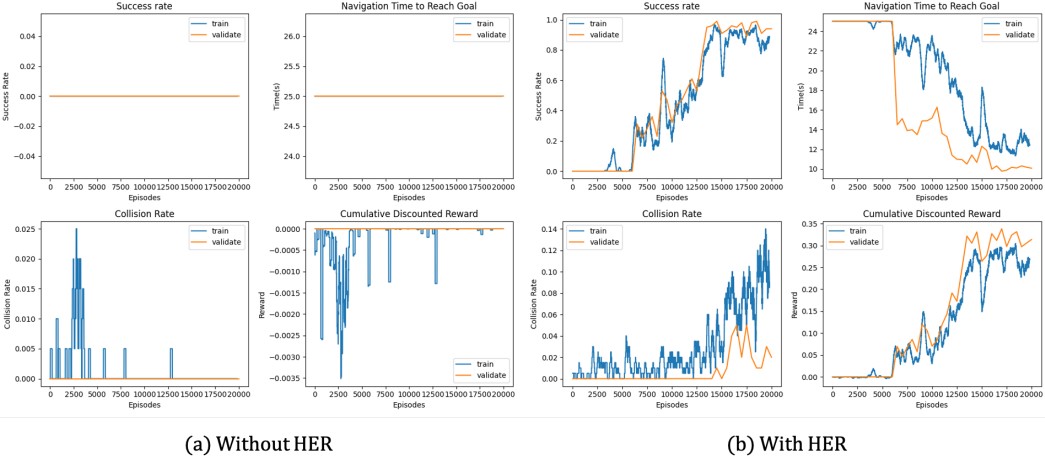

(a) Without HER          (b) With HER

Figure 3: Comparison of the methods with and without HER for navigation in an environment with 1 human.

Secondly, we implement curriculum learning to train the agent to navigate in an environment of 5 humans. Specifically, we initialize the weights of value network using the learned ones in the navigation with 1 human, and train the agent in an environment with 5 humans. Fig. 4 shows the learning

curves of the method using curriculum learning. We can see that with the knowledge of successful navigation with 1 human, the success rate has achieved $49\%$ in validation at the beginning of the training process. After training for 20000 episodes, the success rate converges to about $100\%$ with an almost zero collision rate. Our method with curriculum learning can learn a successful navigation policy in dense crowds without expert demonstrations and achieves a comparative performance with those methods having additional demonstrations. This curriculum learning method can be more applicable in real world navigation problems, as it requires no additional effort for collection of demonstration data.

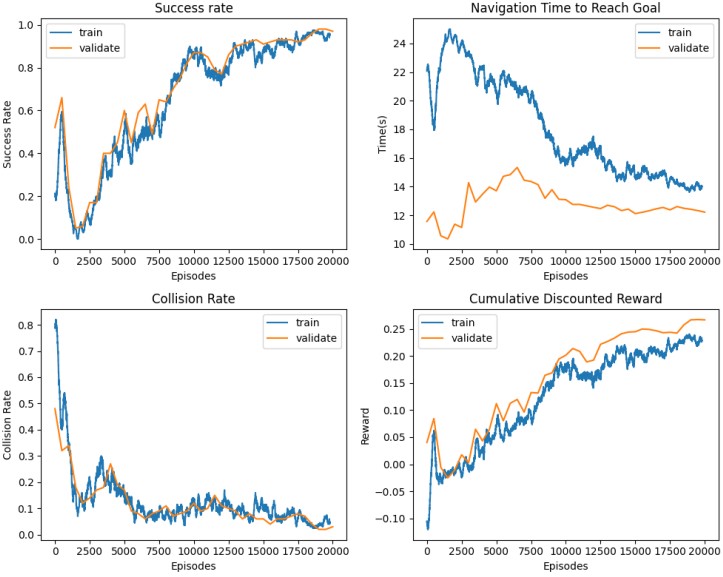

Figure 4: The learning curves of the method using curriculum learning without demonstrations

# 6 Conclusion

In our project, we study the crowd-aware robot navigation problem with deep reinforcement learning. The goal-oriented task is a sparse reward problem. To solve this problem, we first study the performance of different RL algorithms that make use of the expert demonstration data. However, in real world applications, reliable expert demonstrations are hard and expensive to collect. Hence, we aim to develop a more applicable method for the robot navigation without using demonstrations. We propose two possible solutions. One is using well-designed shaped rewards and the other is using curriculum learning with hindsight experience replay technique. While the reward shaping method is limited in high collision rate, we find our latter method can achieve satisfying performance comparable to the methods using additional expert demonstrations.

# References

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
