# OpenReview forum: "Learning Mobile Robot Navigation in the Dense Crowd with Deep Reinforcement Learning"
_CUHK.edu.hk/2021/Course/IERG5350_

### Official Review · AnonReviewer1 · 2020-12-16
**Clear accept as the work shows a comprehensive comparison but lacks remarkable originality and extraordinary significance**

**Rating:** 9
**Confidence:** 5

**Review:**

## Overview
This project aims to learn a human-aware navigation policy by applying an RL method in a robot navigation scenario. It also implements the reward shaping, hindsight experience replay, and curriculum learning techniques for the task. The idea in the paper is how to understand human behavior and take actions under their cooperative rules in a dense crowd socially. The role of expert demonstration becomes what the authors concern about to achieve a desirable navigation performance. The authors state that they use the value function approximation such an RL approach. The overall modification said to us consists of (i) reward shaping, (ii)hindsight experience replay (HER), and (iii) curriculum learning techniques.

In view of the comments below, this reviewer believes that the paper accomplishes a satisfactory and clear illustration for tackling the sparse reward problem. The claimed improvement is worth affirming, especially from a sophisticated navigation point of view. The evaluation is done in simulation. In want of perfection, the originality and significance need to be improved in view of this reviewer. The effectiveness but no comparison with other existing SOTA works is not enough, and thus one would not say a superior performance by the proposed method.

## Quality
The authors conduct the project with high-quality performance comparison experiments and persuasive explanations.
### Pros
+  The experimental results with expert demonstrations part is comprehensive and capable of proving the effectiveness of applying HER and IL to improve three indices.
+ On the other hand, it is impressive to leverage the role of CL with HER in an individual to a crowded environment so that a successful navigation policy in dense crowds without expert demonstrations can be learned. This advances a promising means for navigation in a pedestrian-rich environment.

### Cons
+ A deeper explanation for the function of CL should be given.
+ To achieve adequacy of evidence, please explain the reason why the method with HER in Fig. 1 performs unsatisfactorily compared to the other two methods at the beginning period.

## Clarity
To tackle the sparse reward problem, with and without expert demonstration are considered for the RL agent separately. In both parts, two methods are also implemented and compared, respectively, which presents a progressive and good sense of judgment.
### Pros
+ The structure and organization of this project are clear and meaningful. The derivation of the work is illustrated reasoningly. The expression for conducting an experiment is accuracy and convincing argument. To study the separate contributions of IL and HER to the standard RL algorithm VFA learning, three contrast experiments are conducted under the context of expert demonstrations, paving the way for future comparison.

### Cons
+ For the conclusion part, do not just say what you did, but the effects you got will serve as a better contributor. Namely, the differences between the performances shown by the used RL algorithms should be stated directly.

## Originality
Using the curriculum learning with hindsight experience replay technique in such a scenario is first accomplished while it is not a totally novel method to the RL community.
### Pros
+ It is cheering to see diverse implementations to improve the robotic navigation performance in a crowd scenario, contributing to implications for how to basically deal with exploration and exploitation.

### Cons
+ It is better to refer to some literature that also investigates to solve the sparse reward problem so as to highlight your originality and convince them.


## Significance
This work lays the foundations for developing the methods to improve robotic navigation performance with DRL when facing the sparse reward issue. Researchers who dedicate to cope with densely popular navigation will get instructive ideas from this project.
### Pros
+ Regarding robotic navigation with DRL in a densely popular dynamic environment such a hot topic, the implemented versatile approaches and discoverings are meaningful, especially to deal with label absence situations.

### Cons
+ The applied approaches can only serve as effective means for collision-free navigation, but the performance can not be deemed as having achieved SOTA results. I recommend the authors to refer to [1] from CSAIL Lab of MIT, whose work is based on GA3C.
+ Since the project is about social navigation in a human-aware environment, the significance can be further enhanced if Relational Graph Learning [2] can be considered as a comparison. It will predict state and estimate value to enhance sample efficiency as well, which eventually test a superior performance.

## Other minor issues:
- Citation of references in the text should be quoted as [] not ();
- Improve the writing

> [1] Everett, Michael, Yu Fan Chen, and Jonathan P. How. "Collision Avoidance in Pedestrian-Rich Environments with Deep Reinforcement Learning." arXiv preprint arXiv:1910.11689 (2019).

> [2] Chen, Changan, et al. "Relational graph learning for crowd navigation." arXiv preprint arXiv:1909.13165 (2019).

---

### Official Review · AnonReviewer4 · 2020-12-18
**Very Nice and Informative Paper**

**Rating:** 8
**Confidence:** 5

**Review:**

In general, this paper covers a lot of aspects of robot navigation problem, and touches on the real world situation in which large amount of expert guidance are costly or not possible at all. It compares three methods with expert demonstration, that is, VFA, VFA+IL, VFA+HER, and then compares two methods without human expert, which are reward shaping and curriculum learning. I can follow the logic and progression of this paper clearly and get a well understanding of technical difference and effectiveness of the all the different methods. It covers many contents while being concise about each part.

Significance: Not only does it explores existing well developed robot training techniques (learning with human expertise) but it also puts on efforts to investigate alternative methods to expert demonstration, such that the training method can be more adaptable in real life. This is the most significance aspect of this paper.

Pros: (1) This paper has clear logic, you can find the internal logic between different sections. (2) The figures clearly show useful information and convey a clear idea. (3) This paper is very informative, covering a lots of contents and different methods in real world robot training.

Cons: (1) You should briefly talked about your experiment result in the introduction section. (2) I was not sure whether your agents are both humans and robots or just robots at first, you can make it clear at the beginning of the problem formulation. (3) You can use square bracket for in-line citation to differentiate it from formula.

Most of the problems are minor and grammatical. Technically, it is a very good paper.

---

### Official Review · AnonReviewer2 · 2020-12-20
**An impressive deep RL project**

**Rating:** 9
**Confidence:** 4

**Review:**

Summary:

This report proposes to leverage deep reinforcement learning to human-aware robot navigation. Specifically, the author used the value function approximation algorithm as the basic RL framework. In order to tackle the sparse reward problem of this task, the author further studied various RL techniques, including  Hindsight Experience Replay,  Imitation Learning, Reward Shaping, and Curriculum Learning. These practices are really impressive, and the experimental results also indicate that the proposed RL method can alleviate the sparse reward problem and achieves the navigation task with a low collision rate.

Pros:

1. Well RL project: the author adopted many teciniques to solve the sparse reward problem, which facilitates the RL training.
2. Extensive experiment: the author conducted sufficient experiments to evaluate the effect of each component of the proposed RL framework.
3. Overall, the paper is well-writen.

Cons:

1. Considering the self-contained of the report, it is supposed to append some details about the adopted deep RL architecture.
2. The experiment section is relatively poor-organized. It may due to that the content of this section is too rich to organize :). I think this section should be improved to make it easier to follow, e.g., re-order all sub-sections and present the performance of the naive algorighm at first.

Suggestions:

1. Append some details to make the report self-contained.
2. Improve the experiment section.
3. Use square brackets when citing references, since the original round brackets are easily confused with the notation of equations.